# Development and Validation of Shiga Toxin-Producing *Escherichia coli* Immunodiagnostic Assay

**DOI:** 10.3390/microorganisms7090276

**Published:** 2019-08-21

**Authors:** Miriam A. Silva, Anna Raquel R. Santos, Leticia B. Rocha, Bruna A. Caetano, Thais Mitsunari, Luanda I. Santos, Juliana M. Polatto, Denise S. P. Q. Horton, Beatriz E. C. Guth, Luís Fernando dos Santos, Roxane M. F. Piazza

**Affiliations:** 1Laboratório de Bacteriologia, Instituto Butantan, São Paulo 05503-900, Brazil; 2Departamento de Microbiologia, Imunologia, Parasitologia, Escola Paulista de Medicina Universidade Federal de São Paulo, São Paulo 04023-062, Brazil; 3Centro de Bacteriologia, Instituto Adolfo Lutz, São Paulo 01246-000, Brazil

**Keywords:** STEC, Shiga toxins, antibodies, capture ELISA, latex agglutination test, lateral flow assay

## Abstract

Shiga toxin (Stx)–producing *Escherichia coli* (STEC) and its subgroup enterohemorrhagic *E. coli* are important pathogens involved in diarrhea, which may be complicated by hemorrhagic colitis and hemolytic uremic syndrome, the leading cause of acute renal failure in children. Early diagnosis is essential for clinical management, as an antibiotic treatment in STEC infections is not recommended. Previously obtained antibodies against Stx_1_ and Stx_2_ toxins were employed to evaluate the sensitivity and specificity of the latex Agglutination test (LAT), lateral flow assay (LFA), and capture ELISA (cEIA) for STEC detection. The LAT (mAb Stx_1_ plus mAb stx_2_) showed 99% sensitivity and 97% specificity. Individually, Stx_1_ antibodies showed 95.5% and 94% sensitivity and a specificity of 97% and 99% in the cEIA and LFA assay, respectively. Stx_2_ antibodies showed a sensitivity of 92% in both assays and a specificity of 100% and 98% in the cEIA and LFA assay, respectively. These results allow us to conclude that we have robust tools for the diagnosis of STEC infections.

## 1. Introduction

Among the *E. coli* human pathogens, the Shiga toxin (Stx)–producing *Escherichia coli* (STEC) and its subgroup enterohemorrhagic *E. coli* (EHEC) have gained importance in the three last decades due to their involvement in diarrhea [1], that may be complicated by hemorrhagic colitis (HC) [2] and the hemolytic uremic syndrome (HUS) [3,4], the foremost cause of acute renal failure in children [1] due to action of the two major types of the phage-encoded Stxs, Stx_1_ and/or Stx_2_. HUS is associated more commonly with strains that produce Stx_2_ alone or in combination with Stx_1_ rather than those that produce Stx_1_ only [5,6]. Most cases of the STEC infection are acquired by consuming food of bovine origin; however, other foodstuffs, water, environmental contact, and person-to-person transmission are also important sources [7]. A large fraction of the reported STEC infections is due to *E. coli* O157:H7, the most involved serotype in complicated cases, which often evolve into HUS [8]. However, six serogroups (O26, O45, O103, O111, O121, and O145) account for many cases of non-O157 STEC infections; furthermore the non-O157 serotypes [9,10].

In the Latin American countries, human infections by STEC are endemic in Argentina and are mainly linked to O157 strains. In other Latin American countries STEC (O157 and non-O157) causes sporadic cases of diarrhea, bloody diarrhea, hemolytic anemia and HUS [11,12,13,14,15,16,17]. However, it is essential to point out that the distribution of STEC/EHEC in the gastrointestinal tract of a wide variety of animals indicates the zoonotic character of its infections. The role of different animal species as asymptomatic carriers of STEC/EHEC has been extensively studied in the last years in Brazil. In addition to cattle, which are their most common natural reservoir [18,19] the presence of these pathogens has been identified in the feces of dairy buffaloes [20], sheep [21,22], pigs [23,24], birds [25,26], and fishes [27]. It is noteworthy that some relevant serotypes linked to human infections such as O103:H2 and O157:H7 have been recovered from the feces of sheep and cattle [18,28].

Thus, early diagnosis certainly is fundamental for clinical management of the etiological agent involved in diarrhea; specifically for STEC infections, as the antibiotic treatment is not recommended, since its use may induce the Shiga toxins release, thus allowing its dissemination [29]. Moreover, the diagnosis may be indicative of a likely outbreak, followed by the required measures such as implementation of control and detection of emerging strains [30], thus a key point for therapeutic conduct and consequently to control the disease. The diagnosis of STEC in a routine laboratory is difficult, and only specific virulence factors such as the presence of the Shiga toxin, which is common to all STEC, allows differentiation from other *E. coli* [31].

Immunoserological methods have advantages for clinical laboratories because they significantly reduce the time of analysis, have excellent sensitivity and specificity, and are easy to perform [32]. Despite the availability of commercial immunoassays, such as the ELISA immunoassay (EIA): ProSpecT™ Shiga Toxin *E. coli* (Oxoid Ltd., Basingstoke, UK), Premier^®^ EHEC (Meridian Bioscience, Inc., Cincinnati, OH, USA), Ridascreen^®^ Verotoxin test (R-Biopharm AG, Darmstadt, Germany); Shiga toxin Check™ (TECHLAB, Inc., Blacksburg, VA, USA) and Shiga Toxins, EIA with Reflex to *E coli* O157, Culture (Quest Diagnostics, Inc., Saint Louis, MO, USA); lateral flow assay (LFA): Duopath^®^Verotoxins (Merck & Co., Inc. Palo Alto, CA, USA), ImmunoCard STAT!^®^EHEC (Meridian Bioscience, Inc., Cincinnati, OH, USA), Ridascreen^®^ Quick Verotoxin/O157 (R-Biopharm AG, Darmstadt, Germany) and Shiga toxin Quick Check™ (TECHLAB, Inc., Blacksburg, VA, USA); immunomagnetic separation, such as the RapidCheck^®^Confirm™STEC (Romer Labs Holding, Tulln, Austria) and an optical immunoassay, such as the Biostar OIA Shigatoxin (Inverness Medical Professional Diagnostics, Waltham, MA, USA). These commercial available assays are not implemented in the routine of clinical laboratories of low and middle-income regions’ of developing countries (https://datahelpdesk.worldbank.org/knowledgebase/articles/906519), thus encouraging the present work, i.e., the desire of development of a screening test for the Shiga toxin detection for countries with high incidence, endemic or low information on this infection. Thus, the key of our study is affordability, i.e., to provide the health market, whether a private or public one, an option when analyzing the cost benefit issue (bureaucracy, quality, time to obtain the product and final value). The main project involves two steps: (a) Searching robust tools for the development of the test; (b) will focus on making the use of feces directly, calculating costs and price in the market.

In the present study we limited the search for robust tools, thus essaying the generated polyclonal (pAb) and monoclonal (mAb) antibodies against Stx_1_ and Stx_2_ [33,34,35] and the standardization of three platforms using these antibodies in order to verify their performance. Therefore, these antibodies were employed in an evaluation of the sensitivity and specificity of the immunoserological methods, LAT, cEIA and LFA for detection of Shiga toxin-producing *Escherichia coli* using a collection of bacterial isolates, of which 96 STEC presenting several serotypes and harboring different Stx subtypes and the achieved results indicated that we have robust tools for the the diagnosis of STEC infections.

## 2. Materials and Methods

### 2.1. Bacterial Isolates

We used in this study a collection of 96 Shiga toxin-producing *E. coli* (STEC) strains belonging to different serotypes and *stx* subtypes (Table 1). We also included for the ELISA (EIA) cut-off definition and specificity of the latex agglutination (LA) and lateral flow assay (LFA), 12 typical enteropathogenic *E. coli* (tEPEC) [36,37], 11 atypical enteropathogenic *E. coli* (aEPEC) [38], 45 enterotoxigenic *E. coli* (ETEC) [39,40], nine enteroaggregative *E. coli* (EAEC) [41], eight enteroinvasive *E. coli* (EIEC) [42], 14 diffusely-adherent *E. coli* (DAEC) [42], three fecal *E. coli* negative for DEC virulence factors (NVF *E. coli*), four microbiota *E. coli* isolates and 19 Enterobacteriaceae isolates (*Citrobacter freundii*, *Edwardsiella tarda*, *Enterobacter cloacae*, *Klebsiella pneumoniae*, *K. oxitoca*, *Morganella morganii*, *Proteus mirabilis*, *Providencia spp*., Salmonella Agona, S. Enteritidis, S. Infantis, S. Newport, S. Typhimurium, *Serratia marcescens*, *Shigella boydii*, *S. flexneri*, and *S. sonnei*) from our laboratory collection. The prototype EHEC EDL933 [43] was included in the assay as a positive control for the Stx_1_/Stx_2_ producing strain. To estimate the sample size for sensitivity and specificity of the diagnostic methods herein standardized nomogram determination was done according to Malhotra and Indrayan [44] and Hajian-Tilak [45] based on the studies of the diarrheagenic *E. coli* pathotypes infection prevalence [46,47].

### 2.2. Bacterial Supernatant Preparation

Bacterial supernatants were obtained by sequential bacterial culture in the LB medium for 18 h (1:100), followed by a further 4 h in the EC broth (1:10) containing ciprofloxacin (5 ng/mL) [33] and then lysed with 20% Triton X-100 for capture ELISA (cEIA) and latex agglutination test (LAT) [48] or 200 μg/mL polymyxin sulfate B for lateral flow assay (LFA).

### 2.3. Antibodies

The antibodies employed herein were produced in previous work from our group. Stx_1_ and Stx_2_ polyclonal antibodies (pAbs) were raised in rabbits and characterized elsewhere [33,34]. The generation and characterization of Stx_1_ and Stx_2_ monoclonal antibodies (mAbs) are also described elsewhere [35].

### 2.4. Capture ELISA Immunoassay (cEIA)

Microtiter plates (C96 Polysorp-NUNC) were incubated with 10 µg/mL of Stx_1_-pAb or 25 µg/mL of Stx_2_-pAb in carbonate-bicarbonate-buffered, pH 9.6 at 37 °C for 2 h and then further at 4 °C for 16 h. Phosphate buffered saline (PBS) with bovine serum albumin (BSA) 1% was added as a blocking agent and incubated for 1 h at 37 °C. The supernatant of bacterial cultures were incubated for 1 h at 37 °C. Toxin bound to Stx_1_-pAb or Stx_2_-pAb was then detected with 5 µg/mL of Stx_1_-mAb or Stx_2_-mAb followed by goat anti-mouse IgG peroxidase (Sigma-Aldrich, St Louis, MO, USA) diluted 1:5000 in the blocking solution. Reactions were developed with 0.5 mg/mL O-phenylenediamine (OPD; Sigma Aldrich Co, St Louis, MO, USA) plus 0.5-μL/mL hydrogen peroxide in 0.05 M citrate-phosphate buffer, pH 5.0, in the dark at room temperature. The reactions were interrupted after 15 min by the addition of 50 μL of 1 M HCl. The absorbance was measured at 492 nm in a Multiskan EX ELISA reader (Labsystems, Milford, MA, USA). At each step, the volume added was 100 μL/well, except in the washing and blocking steps, when the volume was 200 μL/well. Between incubations, the plates were washed three times with PBS-Tween 0.05%. All experiments were carried out in duplicate, and results correspond to three independent experiments.

### 2.5. Latex Agglutination Test (LAT)

The beads were coupled with 100 µg of Stx_1_ mAb for LAT-Stx^1^ or 100 µg of Stx_2_ mAb for LAT-Stx_2_. For detection of Stx without subtype discrimination, the beads were coupled with Stx_1_ and Stx_2_ mAbs (1:1). The principle used was the nucleophilic addition to aldehyde group with amines. The glutaraldehyde was used as a spacer *arm* between the bead and mAbs. Briefly, the polybeads amino microsphere in a 2.5% aqueous suspension (1 µm diameter–Polyscience, Warrington, PA, USA) were washed three times with PBS and incubated with 8% glutaraldehyde in the PBS at room temperature for 4 h [48]. Next, 50 µg of Stx_1_ and 50 µg of Stx_2_ mAbs were added and the mixture incubated at room temperature for 16–18 h for coupling, followed by further incubation in the presence of 0.2 M ethanolamine and BSA. Both incubations were with gentle mixing at room temperature for 60 min. Between incubations, the coated beads were washed and centrifuged (7200× *g*) for 6 min. After the last washing procedure, the pellet was resuspended in the storage buffer (Polyscience, Warrington, PA, USA) and kept at 4 °C.

For LAT, bacterial lysate were obtained by sequential bacterial growth in the LB medium for 18 h (1:100), followed by a further 4 h in the EC broth (1:10) containing ciprofloxacin (5 ng/mL) and then, lysed with 20% Triton X-100 during 1 h and centrifuged (14,500× *g*) for 15 min. The assay was performed on a slide glass using 20 µL of bacterial lysate and 20 µL of latex beads coupled to Stx_1_ plus Stx_2_ mAbs, and checking for agglutination between 1–2 min of gentle mixing.

### 2.6. Lateral Flow Assay (LFA)

The Stx_1_ or Stx_2_ pAb rabbit sera were used as a capture antibody conjugated to colloidal gold particles. The protocol for conjugation of the pAbs with colloidal gold was performed according to Oliver (2010) [49], with some modifications. Briefly: 0.01% colloidal gold solution (20 nm diameter particles) (BBInternational, Cardiff, England) was previously adjusted to pH 9.0 with 0.1 M potassium carbonate solution (K_2_CO_3_). First, 1 mg of each of the pAb was resuspended in 1 mL of 0.2 M borate buffer (0.2 M sodium borate, 0.15 M NaCl, pH 9.0) and dialyzed against 2 mM borate buffer (2 mM sodium borate, 1.5 mM NaCl, pH 9.0) at room temperature for 2 h.

The amount of antibody required to stabilize colloidal gold was determined as follows: 100 μL of colloidal gold was added in microtubes containing 10 μL of serially diluted previously dialyzed pAb. After 10 min, 11 μL of 10% sodium chloride (NaCl) was added to each tube. The amount of antibody sufficient to stabilize the gold was the dilution in which the solution did not change color, i.e., an insufficient amount of antibody altered the coloration of the solution from red to blue [49].

After determination of the optimal ratio between the antibody and colloidal gold made in the previous step, a total volume of 10 mL of solution was obtained, which was kept under stirring at room temperature for 30 min. To block the reaction, a 10% BSA solution (10% BSA in 0.02 M borate buffer, pH 9.0) was added in sufficient volume to the final concentration of 1%. After incubation at room temperature for 30 min, the solution was centrifuged at 11,000× *g* for 20 min at room temperature and the supernatant discarded. The pellet was then resuspended in 2 mL of a 2% BSA solution (2% BSA in 0.01 M borate buffer) and centrifuged at 11,000× *g* for 10 min at room temperature. Finally, the pellet was resuspended in 1 mL of the storage buffer (3% BSA, 3% sucrose, 0.01 M sodium borate, and 0.05% sodium azide, pH 7.5) and the pAbs conjugated to colloidal gold (pAb-Au) were stored at 4 °C. Finally, 600 μL of pAb-Au was applied at 30 cm in the dried glass fiber in a desiccator whose relative humidity is 20% for 24 h.

The Stx_1_ (3 mg/mL) or Stx_2_ (4 mg/mL) mAbs were used for detection, and they were applied directly to the nitrocellulose membrane (Millipore HF180 NM) as the test line. As a control line of the LFA, the goat anti-rabbit IgG antibody was applied above the test line. The LFA tests were prepared on a large scale on a semi-automatic platform, consisting of a Matrix 1600 Reagent Dispensing Module (applies the antibodies in NM), Matrix 2210 Universal Laminator Module (mounts/overlaps all test membranes), and Matrix 2360 Programmable Shear (cuts LFA test strips), all obtained from Kinematic Automation, Inc. (Sonora, CA, USA). The strips were dried at room temperature in a desiccator whose UR is 20% for 24 h. The treatment of the sample pad portion was made by buffer containing 1% BSA, 0.25% Tween-20, and sodium azide.

For LFA, the bacterial supernatant were obtained by sequential bacterial growth in the LB medium for 18 h (1:100), followed by a further 4 h in the EC broth (1:10) containing ciprofloxacin (5 ng/mL) and then, lysed with 200 μg/mL polymyxin sulfate B during 1 h and centrifuged (14,500× *g*) for 15 min. Supernatant were kept at −20 °C until test analyzes.

### 2.7. Stx Subtyping

Stx subtyping was performed by PCR using the primers and amplification conditions as previously described [50].

### 2.8. Statistical Analyses

The Vero cell assay (VCA) was employed as the gold standard method for the Stx production [51]. Additionally, we differentiated the toxin subtype by PCR for *stx_1_* and or *stx_2_* in order to evaluate the sensitivity and specificity of Stx_1_ or Stx_2_ antibodies. The absorbance values from the duplicates of three independent experiments from Stx-positive and Stx-negative isolates after reaction with Stx_1_ or Stx_2_ antibodies were analyzed by GraphPrism 5.01, using the Student’s *t*-test and two-way ANOVA. The differences were considered statistically significant when *p* ≥ 0.05. The receiver operating characteristic (ROC) curve was employed for determining the calculation of the ELISA’s cut-off as well as providing the sensitivity and specificity report. Furthermore, the Cohen’s Kappa statistic was employed to test the interrater reliability [52]. 

## 3. Results

### 3.1. Presence and Production of Stx_1_ and Stx_2_

The production of Stx was analyzed in the bacterial collection (221 isolates) employing the gold standard Vero cell assay (VCA). The 96 STEC were confirmed as Stx producers, and the 125 other enterobacterial strains were Stx negative. In addition, the *stx* subtype was defined by PCR in the 96 STEC strains, in which 47 were *stx_1_* and 28 were *stx_2_* and 21 presented both genes (Table 1).

### 3.2. Validation of Diagnostic Immunoassays

In order to test the bacterial supernatants for each different assay, bacterial cultures were prepared differently. Thus, for capture ELISA (cEIA) and latex agglutination test (LAT) bacterial culture was lysed with Triton X-100 or polymyxin sulfate B for the lateral flow assay (LFA), since the detergent presence impaired the sample flow in LFA. 

The bacterial collection mentioned above was analyzed using three different immunoassays employing Stx_1_ and Stx_2_ polyclonal and monoclonal antibodies raised in-house [33,34,35] in order to observe their performance in the screening assay (LAT and/or LFA) and/or confirmatory assay (cELISA). These analyses allowed calculating the assay parameters as predictive value (PV) for the positive (PPV) and negative (NPV) samples, the accuracy (A), the sensitivity (Se) and specificity (Sp). In addition, the kappa concordance index (κ) was evaluated using the *p* value < 0.001.

The analysis of the Stx_1_ detection in 68 strains (positive samples) and 153 strains (negative samples) by the three methods revealed for cEIA was PPV = 94%, NPV = 98%, A = 97%, Se = 95.5%, Sp = 97% and κ = 0.957. For LAT: PPV of 82%, NPV of 97% and A of 92%, thus the Se and Sp was 94% and 91%, respectively, and κ = 0.846. The parameters observed for LFA: 97% for PPV, 97% for NPV, 97% for A, 94% for Se, 99% for Sp and κ = 0.925 (Figure 1 and Figure 3).

Concerning the detection of Stx_2_ by 49 strains (positive samples) and 171 strains (negative samples) by the three methods revealed for cEIA: PPV = 100%, NPV = 98%, A = 98%, Se = 92%, Sp = 100% and κ = 0.933. These values for LAT: PPV of 81%, NPV of 99% and A of 94%, thus the Se and Sp was 96% and 93.5%, respectively, and κ = 0.829. The parameters observed for LFA: 94% for PPV, 98% for NPV, 97% for A, 92% for Se, 98% for Sp and κ = 0.883 (Figure 2 and Figure 3).

Comparing the values of the three immunoassays for Stx_1_ and Stx_2_, LAT employing mAbs individually coupled to latex particles always showed lower predictive values than LFA and cEIA. Thus, we decided to combine both monoclonal antibodies (Stx_1_ and Stx_2_) in order to detect Stx without distinction between the toxin types. We achieved the following values: PPV of 96%, NPV of 99% and A of 98%, thus the Se and Sp was 99% and 97%, respectively and κ = 0.945. Therefore, we observed five false positives ((three strains of DAEC (190 and 203), one EAEC (BA1348) and one EIEC (167(48)) and one false negative, the test was not able to detect one O157:H7 strain (Figure 4 and Figure 5). 

In the cEIA, the Stx_1_ Abs (mAb and pAb) were able to recognize the Stx_1_ producers, showing 95.5% of sensitivity and 97% of specificity and an A_492nm_ of 0.195 cut off (Figure 6A) and Stx_2_ Abs (mAb and pAb) were able to recognize the Stx_2_ producing strains, showing 92% of sensitivity and 100% of specificity and an A_492nm_ of 0.1205 cut off (Figure 6B).

## 4. Discussion

Different protocols for detection of Shiga toxin-producing strains either in the feces of infected patients or contaminated food have already been described for routine diagnosis [53,54,55]. Currently, the PCR for *stx* has been employed in reference diagnosis centers and some laboratories of upper-middle-income regions of developing countries using bacterial confluent growth zones or sorbitol-fermenting and non-fermenting colonies taken from MacConkey sorbitol agar (SMAC plates) [47,53]. Nevertheless, the gold standard for Stx detection is the evaluation of the cytotoxicity of bacterial culture supernatants for eukaryotic cells (VCA) [3,51].

Thus encouraging the present work, i.e., the request of a development of a screening test for the Shiga toxin detection for countries with high incidence, endemic or low information on this infection. Thus, Stx_1_ and Stx_2_ pAbs and mAbs were generated in previous studies [33,34,35]; and different formats of immunoassays, employing these antibodies were developed and standardized. Herein, we evaluated the sensitivity and specificity of LAT, cEIA, and LFA employing VCA as gold standard.

An important point is which protocol should be used for toxin production and secretion. Usually, Stx_1_ is secreted into the medium [56], whereas Stx_2_ has been shown to be either periplasmatic [56] or liberated inside vesicles [57] therefore not secreted into the medium. In a former work of our group we have established a protocol for in vitro cultivation by sequential bacterial culture in LB medium for 18 h (1:100), followed by a further 4 h in the EC broth (1:100), and in order to enhance toxins secretion we added ciprofloxacin (5 ng/mL) [33]. Herein, we established a 1:10 dilution from the LB to the EC broth, in turn; to better stimulate toxin production, since in 1:100 dilutions the very low-producers strains were not detected. Another critical point we established is the different procedures for toxins secretion according to the assay requirements. Thus, bacterial supernatants were prepared differently; bacterial culture was lysed with Triton X-100 for cEIA and LAT or polymyxin sulfate B for LFA. Usually, the commercial available tests also employ different procedures for toxin secretion, except for the Ridascreen^®^ Verotoxin test and Ridascreen^®^ Quick Verotoxin/O157, that mention the use of mitomycin C as an inductor for the formation of Shiga toxins, the others refer generically to the diluent.

The sensitivity and specificity of each tested assay reached percentages as the commercial ones (pertaining the datasheet of each company), therefore encouraging results; concerning cEIA-Stx_1_ 95.5 and 97%, respectively and cEIA-Stx_2_ 92 and 100%, respectively, comparable to ProSpecT™ Shiga Toxin *E. coli* (92.3 and 99.6%); Premier^®^ EHEC (100 and 97.9%); Ridascreen^®^ Verotoxin test (93.9 and 96.1%) and Shiga toxin check™ (97.1 and 99.7%). For LFA-Stx_1_ (94 and 99%) and LFA-Stx_2_ (92 and 98%) for sensitivity and specificity, respectively; comparable to Duopath^®^Verotoxins (100 and 99.6%); ImmunoCard STAT!^®^EHEC (92.3 and 98.7%); Ridascreen^®^ Quick Verotoxin/O157 (85 and 98.7%) and Shiga toxin Quick Check™ Stx_1_ (100 and 99.5%) and Stx_2_ (95.7 and 99.9%).

Regarding the sensitivity and specificity of the latex agglutination assay, LAT-Stx_1_ (94 and 91%); LAT-Stx_2_ (96 and 93.5%), respectively, but when both mAbs were combined these results reached to 99 and 97%, respectively. Our results cannot be compared to those of the reverse passive latex agglutination kits (VTEC-RPLA toxin detection and VTECRPLA “Seiken”, Japan) since they were no longer commercially accessible. Early diagnosis of diarrhea is the key to therapeutic behavior, accordingly, for a diagnostic assessment method we may employed RALT for EspB (97% sensitivity and 98% specificity) [48] and LAT-Stx (99% sensitivity and 97% specificity), so we may define in a short term if the diarrhea was due to EPEC/EHEC or STEC. Only one false negative occurred and five false-positives, in fact, among them, one strain initially described as the adherent diffuse *E. coli* (DAEC) by the *daaC* probe [42]. Herein it is a real positive, detected by PCR as Stx_2_e, but a non-producer Stx by VCA. Moreover, no cross-reaction was observed neither with *E. coli* negative for the DEC virulence factors nor with the microbiota strains and among the enterobacterial species.

Due to the feasibility of LFA, the LFA-Stx_1_ and LFA-Stx_2_ we consider that it can be used as a rapid test, such as described elsewhere [58,59], since it did not require expensive equipment or trained personnel to interpret the results. Unlike ELISA or PCR methods, colloidal gold technology can be used for point-of-care applications and screening as they require the only assessment of red colored lines for the end-point detection. Both standardized methods are reproducible, fast, easy to perform, showing high sensitivity in detecting Stx.

The same features were observed in cEIA, even detecting Stx in low-producing isolates, but, cEIA can be used as a diagnostic helper or confirmatory since a spectrophotometer is necessary for the absorbance’s reading. Currently, these assessment immunoassays are under technology transfer to a Brazilian start-up in order to validate those employing fecal samples and the commercial availability of them.

## 5. Conclusions

The standardized tests can be used not only in reference laboratories but also mainly in clinical laboratories and hospitals, given the importance of diagnosis for appropriate patient treatment and the prevention of outbreaks and contamination by STEC. The comparative analysis of cEIA, LAT and LFA allows the conclusion that we have robust tools for STEC diagnosis infections. Assessing all our data, including the rapidity of feedback to the patient, feasibility, and accuracy of the test, we can also conclude that when the Stx_1_ and Stx_2_ mAbs were coupled, better performance was observed for a screening test for the Shiga toxin detection.

## Figures and Tables

**Figure 1 microorganisms-07-00276-f001:**
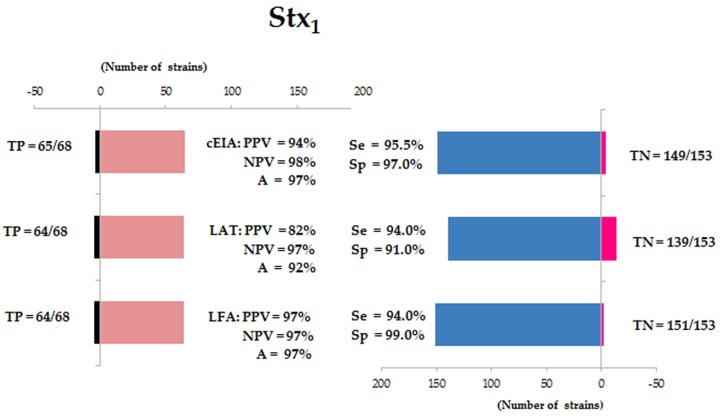
Analysis of Stx_1_ detection by capture ELISA (cEIA), latex agglutination test (LAT) and the lateral flow assay (LFA). TP = true positive strains; TN = true negative strains; PPV = positive predictive value; NPV = negative predictive value; A = accuracy; Se = sensitivity and Sp = specificity.

**Figure 2 microorganisms-07-00276-f002:**
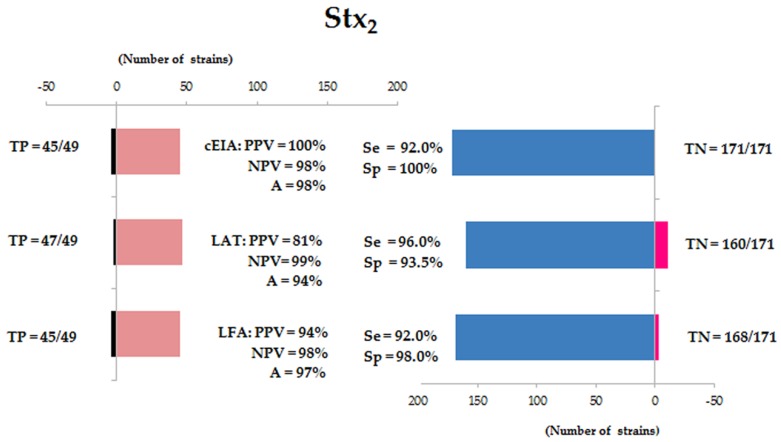
Analysis of Stx_2_ detection by capture ELISA (cEIA), latex agglutination test (LAT) and lateral flow assay (LFA). TP = true positive strains; TN = true negative strains; PPV = predictive positive value; NPV = negative predictive value; A = accuracy; Se = sensitivity and Sp = specificity.

**Figure 3 microorganisms-07-00276-f003:**
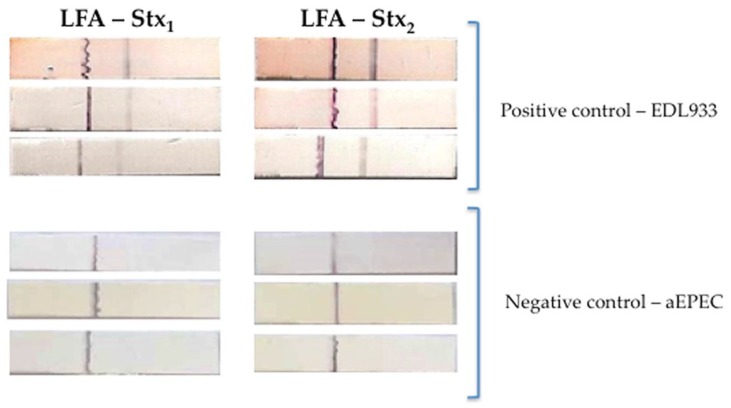
Representative image of positive (EDL933 – O157:H7) and negative (atypical EPEC–aEPEC- O127:H7) controls strains in a lateral flow assay (LFA) employing Stx_1_ or Stx_2_ antibodies. Triplicates of each experiment.

**Figure 4 microorganisms-07-00276-f004:**
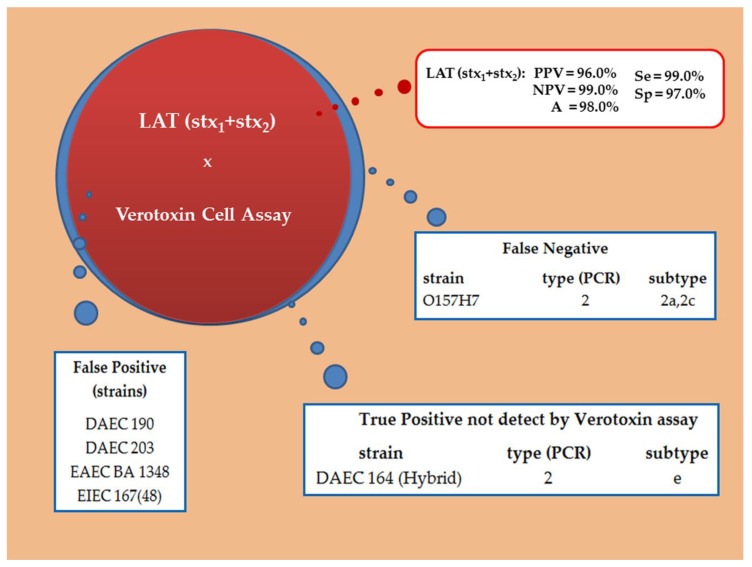
Venn diagram of latex agglutination test (LAT) employing Stx_1_ mAb combined with Stx_2_ mAb; PPV = positive predictive value; NPV = negative predictive value; A = accuracy; Se = sensitivity and Sp= specificity. Blue = VCA results; Red = LAT results.

**Figure 5 microorganisms-07-00276-f005:**
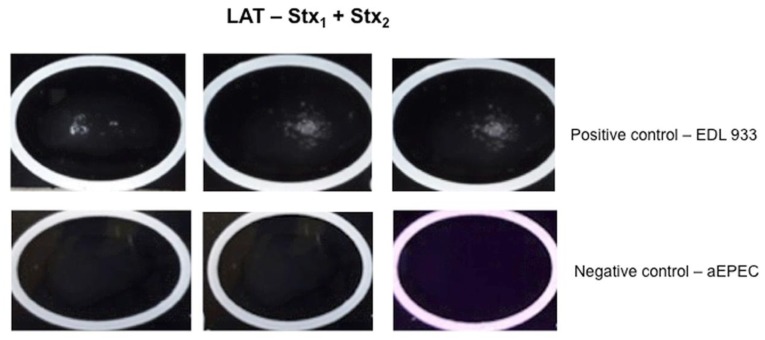
Representative image of positive (EDL 933 – O157:H7) and negative (atypical EPEC–aEPEC – O127:H7) controls strains in a latex agglutination test (LAT) employing Stx_1_ and Stx_2_ antibodies. Triplicates of each experiment.

**Figure 6 microorganisms-07-00276-f006:**
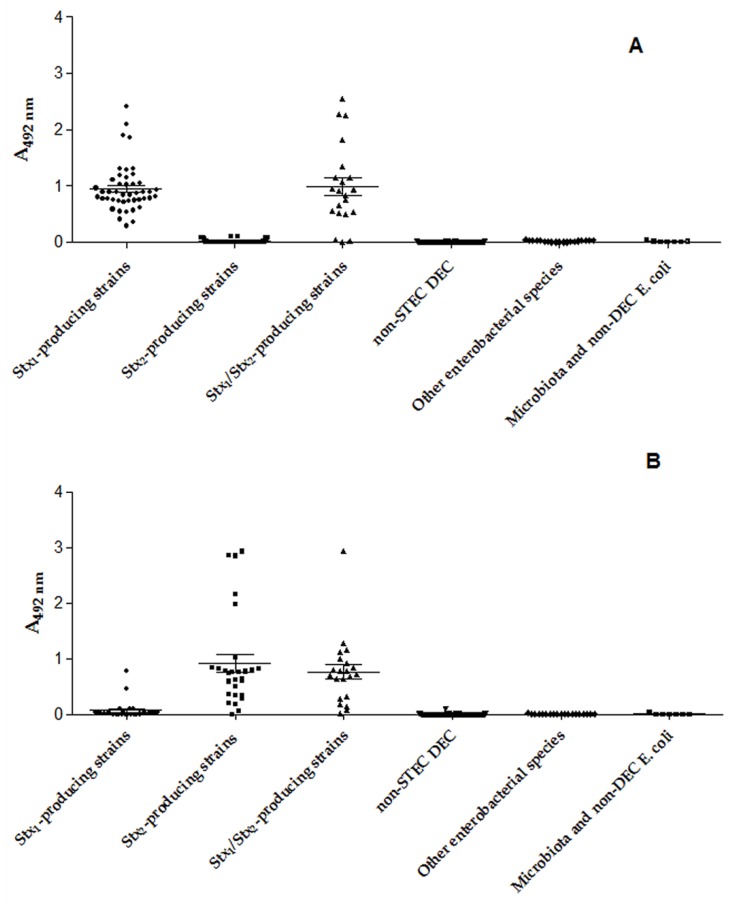
Capture ELISA (cEIA) analysis: (**A**) Employing Stx_1_ pAb and mAb; (**B**) Employing Stx_2_ pAb and mAb. The optical densities obtained for the isolates reacted with Stx_1_, or Stx_2_ pAb and mAb were analyzed by GraphPrism 5.01, using the Student’s t-test and two-way ANOVA. The differences were considered statistically significant when *p* ≤ 0.05.

**Table 1 microorganisms-07-00276-t001:** Shiga toxin-producing *E. coli* (STEC) strains immunoassay results.

Strain ID	Serotype	Gene Presence *stx_1_/stx_2_*	*stx* Subtype	cEIA (Stx_1_)	cEIA (Stx_2_)	LAT (Stx_1_ + Stx_2_)	LFA (Stx_1_)	LFA (Stx_2_)
IAL6189	O24:H4	1	1a	1.149	0.000	+	+	−
IAL6206	O24:H4	1	1a	0.717	0.000	+	+	+
IAL6163	O26:H11	1	1a	0.810	0.000	+	+	−
IAL6162	O26:H11	1	1a	0.777	0.000	+	+	−
H30	O26:H11	1	1a	1.207	0.003	+	+	−
H19	O26:H11	1	1a	1.109	0.000	+	+	−
199	O26:H11	1	1a	0.892	0.000	+	+	−
3529	O26:H11	1	1a	0.772	0.000	+	+	−
EPM16	O26:H11	1	1a	1.019	0.006	+	+	−
BA4123	O26:H11	1	1a	0.853	0.000	+	+	−
D360-4-1	O26:H11	1	1a	0.832	0.000	+	+	−
1557-77	O26:H11	1	1a	2.416	0.061	+	+	−
CL5	O26:H12	1	1a	0.760	0.000	+	+	−
EPM5	O55:H19	1	1a	0.896	0.008	+	+	−
IAL6174	O71:H8	1	1a	0.767	0.000	+	+	−
IAL6290	O76:H19	1	1c	0.545	0.000	+	+	−
IAL6173	O91:H14	1	1a	0.288	0.013	+	+	−
IAL6186	O103:H^−^	1	1a	1.027	0.016	+	+	−
IAL6175	O111:H^+^	1	1a	1.900	0.100	+	+	−
IAL6191	O111:H^+^	1	1a	0.739	0.000	+	+	−
IAL6184	O111:H8	1	1a	0.857	0.007	+	+	−
IAL6177	O111:H8	1	1a	2.093	0.088	+	+	−
IAL6200	O111:H8	1	1a	0.741	0.000	+	+	−
EPM20	O111:H8	1	1a	0.539	0.000	+	+	−
IAL6183	O111:H8	1	1a	0.893	0.000	+	+	−
EPM23	O111:H8	1	1a	0.831	0.000	+	+	−
IAL6187	O111:H11	1	1a	0.783	0.011	+	+	−
IAL6178	O111:HNM	1	1a	0.354	0.000	+	+	−
EPM26	O111:HNM	1	1a	0.931	0.000	+	+	−
EPM27	O111:HNM	1	1a	1.053	0.000	+	+	−
EPM017	O112:H2	1	1c	0.610	0.000	+	+	−
EPM11	O118:H16	1	1a	0.869	0.000	+	+	−
IAL6196	O118:H16	1	1a	0.801	0.000	+	+	−
IAL6188	O118:H16	1	1a	0.566	0.009	+	+	−
IAL6171	O123:H^−^	1	1a	0.897	0.000	+	+	−
IAL6181	O123:H2	1	1a	1.857	0.098	+	+	−
IAL6180	O123:H2	1	1a	0.888	0.010	+	+	−
IAL6197	O123:HNM	1	1a	0.749	0.000	+	+	−
IAL6192	O153:H21	1	1a	0.733	0.000	+	+	−
82	O157:H7	1	1a	0.963	0.000	+	+	−
3299-85	O157:H7	1	1a	1.032	0.078	+	+	−
46240	O157:H7	1	1NT	1.188	0.000	+	+	−
3077-88	O157:H7	1	1a	1.281	0.076	+	+	−
C7-88	O157:H7	1	1a	1.312	0.006	+	+	−
EPM01	ONT:H8	1	1c	0.584	0.000	+	−	−
184332	OR:H19	1	1a	0.408	0.000	+	+	−
BA597	OR:NM	1	1a	1.312	0.000	+	+	−
IAL6176	O8:H19	2	2a + 2d	0.017	0.181	+	−	−
01-9582-01	O39:HR	2	2f	0.018	2.929	+	−	+
IALEc1054/05	O91:H21	2	2a + 2c	0.025	0.829	+	−	+
IAL6201	O100:H^−^	2	2e	0.028	0.602	+	−	+
EPM82	O112:H21	2	2c	0.020	0.747	+	−	+
IALEc226/04	O113:H21	2	2a	0.100	0.585	+	−	+
IALEc678/04	O113:H21	2	2a	0.024	2.858	+	−	+
IALEc603/04	O141:H49	2	2a	0.013	2.165	+	−	+
IAL6182	O153:H28	2	2NT	0.000	0.202	+	−	+
IALEc1167/05	O157:H^−^	2	2a + 2c + 2e	0.768	0.339	+	+	+
IALEc703/04	O157:H^−^	2	2a + 2d	0.017	0.737	+	−	+
IAL6193	O157:H7	2	2a + 2c	0.000	1.022	+	−	+
IAL6207	O157:H7	2	2a + 2c	0.000	0.621	+	−	+
IAL6179	O157:H7	2	2a + 2c	0.000	0.268	−	−	+
IAL6202	O157:H7	2	2a + 2c	0.054	0.764	+	−	+
EPM1	O157:H7	2	2a + 2c	0.092	2.847	+	−	+
EPM2	O157:H7	2	2a + 2c	0.014	0.776	+	−	+
EPM03	O172:NM	2	2a	0.000	0.637	+	−	+
IAL6199	O177:H^−^	2	2c	0.000	0.752	+	−	+
IAL6172	O178:H19	2	2c	0.102	0.810	+	+	+
IALEc170/04	ONT:H7	2	2a + 2f	0.463	0.000	+	+	−
EPM59	ONT:H16	2	2d	0.000	0.349	+	−	+
EPM022	ONT:H16	2	2b	0.012	0.058	+	−	−
IALEc157/05	ONT:H23	2	2c + 2d	0.027	0.332	+	−	+
IAL6195	ONT:H46	2	2a + 2d	0.000	0.489	+	−	+
BA1132	ONT:H49	2	2a + 2c + 2d	0.010	0.809	+	−	+
BA1189	ONT:H49	2	2a + 2d	0.016	0.792	+	−	+
IAL6198	OR:H^−^	2	2c	0.005	1.985	+	−	+
EPM79	O22:H16	1/2	1a + 2c + 2d	0.022	0.633	+	−	+
IALEc515/05	O43:H2	1/2	1NT + 2NT	0.558	0.176	+	+	+
BA3003	O48:H7	1/2	1a + 2a	0.651	0.638	+	+	+
IALEc169/04	O74:H25	1/2	1a + 2c	2.267	1.149	+	+	+
EPM036	O75:H8	1/2	1c + 2b	1.153	0.005	+	+	−
IAL6208	O75:H14	1/2	1c + 2NT	1.819	0.141	+	+	+
IALEc617/04	O84:HNM	1/2	1NT + 2NT	0.049	0.794	+	−	+
EPM50	O87:H16	1/2	1NT + 2b	0.008	0.071	+	−	+
EPM4	O93:H19	1/2	1a + 2d	2.538	0.703	+	+	+
EPM44	O98:H4	1/2	1a + 2NT	0.910	0.778	+	+	+
EPM53	O98:H17	1/2	1a + 2a + 2c	0.747	0.781	+	+	+
EPM55	O98:H17	1/2	1a + 2a + 2c	0.839	0.712	+	+	+
EPM9	O103:H2	1/2	1a + 2c	0.940	0.912	+	+	+
EPM66	O105:H18	1/2	1a + 2a + 2b	1.069	0.839	+	+	+
EPM055	O146:H21	1/2	1a + 2a + 2b	1.144	0.674	+	+	+
3104-88	O157:H7	1/2	1a + 2a	0.502	1.111	+	+	+
EDL933	O157:H7	1/2	1a + 2a	1.348	1.277	+	+	+
EPM45	O181:H4	1/2	1a + 2a	0.941	0.274	+	+	+
IALEc161/04	ONT:H18	1/2	1a + 2a + 2c	2.252	2.927	+	+	+
EPM81	ONT:H38	1/2	1NT + 2a	0.517	0.990	+	+	+
IALEc630/04	ONT:H46	1/2	1a + 2f	0.532	0.323	+	+	+

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
