# Peer review of "Development and Validation of Shiga Toxin-Producing Escherichia coli Immunodiagnostic Assay"

_microorganisms, 2019, doi:10.3390/microorganisms7090276_

Round 1
Reviewer 1 Report
1. Title (L.2-6). If I have accurately captured the authors’ intent, I suggest that the title be changed to: “Development and validation of Shiga toxin-producing Escherichia coli immunodiagnostic assays”. (The title is too long, and too detailed. Affordability is the key; however, this study does not strictly demonstrate the difference in cost from current commercial assays. The last paragraph mentions it is unknown what the actual costs will be. Further, anyone, even those in countries with higher incomes, would benefit from the availability of less expensive assays, right? The authors’ specific intent to overcome the lack of availability in lower income areas along with other details can be provided in the Abstract and Discussion.)
2. Abstract (L.23-36). Although it is explained somewhat in the last paragraph of the Discussion (e.g., reduced cost of equipment needed), it is not exactly clear how this study makes immunoassays more affordable to low- and middle-income regions of the world. (Will the mAb and pAb reagents be made available at no cost, or at limited cost?) Please edit the Abstract and Discussion (see below) accordingly.
3. L.50. Delete “that is”
4. L.50-51. Change “which usually evolving to HUS” to “which often evolve into HUS”
5. L.52. Insert “many cases of” between “for” and “non-O157”
6. L.52. The phrase “also, attention should be given to these non-O157 serotypes” is unclear. What kind of attention?
7. L.56. Change “represents” to “causes”
8. L.66. Delete “the”
9. L.67. Change “induces Shiga toxins secretion,” to “may induce Shiga toxin release,”
10. L.78. What is meant by the words “with reflex”?
11. L.82. Insert a comma and “such” after “separation” (i.e., “…separation, such as the Rapid…”)
12. L.83. Insert a comma and “such” after “immunoassay” (i.e., “…immunoassay, such as the Biostar…”)
13. L.104. Change “Klebsiella oxitoca” to “K. oxytoca” (since Klebsiella was spelled out already with pneumoniae and the species is misspelled.)
14. L.104-105. agona, enteritidis, infantis and others are serotypes, not species (they are all serotypes within the species enterica, and by convention should begin with a capital letter and not be italicized. Also, “Salmonella” can be abbreviated as “S.” after it is first spelled out. Hence: “…Salmonella Agona, S. Enteritidis, S. Infantis, S. Newport, S. Typhimurium, …”
15. L.106. Abbreviate “Shigella” as “S.” after it is first spelled out. Hence. “…Shigella boydii, S. flexneri, and S. sonnei)…”
16. L.111. Change “infections” to “infection”
17. L.120-121. Change “…are described in [35].” to “…are also described elsewhere [35].”
18. L.124. Insert “at” between “further” and “4”
19. L.125. Change “Buffered” to “buffered”
20. L.152. Change “E. coli broth” to “EC Broth” and list manufacturer name and location in parentheses.
21. L.168. This reference should be no. 58, not 57. Please review the entire manuscript to make sure there are no more reference errors.
22. L.179. Please spell out UR before abbreviating.
23. L.194-195. Change “…conditions described by [50].” to “conditions as previously described [50].”
24. L.197. Insert “The” before “Vero” and “the” before “gold” in this sentence.
25. L.198. Change “differentiate” to “differentiated”
26. L.198 and throughout the manuscript. Change “stx1” to “stx1” and “stx2” to “stx2”
27. L.205. The meaning of “inter tests” is unclear. Please explain or reword.
28. L.207. Please deleted “toxins” (Stx1 and Stx2 are already understood to be toxins)
29. L.208. Please delete “studied”
30. L.211. Change “…; besides,…” to “…, and…”
31. L.212. Table 1 title. Delete “feature” and change “immunoassays” to “immunoassay”.
32. L.214. Insert a comma after “assay”
33. L.215 and elsewhere in the manuscript. Change “Latex” to “latex”
34. L.216 and elsewhere in the manuscript. Change “Lateral” to “lateral”
35. L.221. Change “assays” to “assay”
36. L.223. Change “Besides” to “In addition” and delete “also”
37. Figures 1 and 2 and their legends. In the figure change “(Number of the strains)” to “(Number of strains).” In the legend, change “True Positive Strains” to “true positive strains”, “True Negative Strains” to “true negative strains”, “Predictive Positive Value” to “positive predictive value”, “Negative Predictive Value” to “negative predictive value”, “Accuracy” to “accuracy”, “Sensitivity” to “sensitivity”, and “Specificity” to “specificity”.
38. Figure 3. In the figure, change “Non detect by Verotoxin” to “Not detected by Verotoxin assay”. In the legend (L.255-256), change “Predictive Positive Value” to “positive predictive value”, “Negative Predictive Value” to “negative predictive value”, “Accuracy” to “accuracy”, “Sensitivity” to “sensitivity”, and “Specificity” to “specificity”.
39. Figure 4. On the x-axis, change the following legends: “DEC non-STEC” to “non-STEC DEC”, “Enterobacterial” to “enterobacterial”, “Microbiota and E. coli non-DEC” to “Microbiota and non-DEC E. coli”
40. L.270. Delete “gene”
41. L272. Change “sorbitol fermenting” to “sorbitol-fermenting”; change “Sorbitol” to “sorbitol”
42. L274. Change “to” to “for”
43. L.284. Change “…tests as …” to “…tests, such as…”
44. L.285. Change “…test as…” to “…tests, such as…”
45. L.287. Delete “Regarding the”, begin the sentence with “Lateral”, and insert a comma after “(LFA)”
46. Discussion. Will the reagents (mAb and pAb) be made available at a low cost, or only newly developed assays? How do the authors know for sure the cost will be less than commercial assays?
47. Change “commercially” to “commercial”
Author Response
We would like to acknowledge the reviewer comments
Comments and Suggestions for Authors
Title (L.2-6). If I have accurately captured the authors’ intent, I suggest that the title be changed to: “Development and validation of Shiga toxin-producing Escherichia coli immunodiagnostic assays”. (The title is too long, and too detailed. Affordability is the key; however, this study does not strictly demonstrate the difference in cost from current commercial assays. The last paragraph mentions it is unknown what the actual costs will be. Further, anyone, even those in countries with higher incomes, would benefit from the availability of less expensive assays, right? The authors’ specific intent to overcome the lack of availability in lower income areas along with other details can be provided in the Abstract and Discussion.)We appreciate the suggestions; we changed the tittle as suggested. As well as the details in the abstract, introduction and discussion. All modifications are highlighted.
Abstract (L.23-36). Although it is explained somewhat in the last paragraph of the Discussion (e.g., reduced cost of equipment needed), it is not exactly clear how this study makes immunoassays more affordable to low- and middle-income regions of the world. (Will the mAb and pAb reagents be made available at no cost, or at limited cost?) Please edit the Abstract and Discussion (see below) accordingly.
The abstract and discussion are now edited according your suggestions.
We checked all the grammar corrections and spellings as indicated below. The references were carefully checked and modified. All modifications are highlighted.
L.50. Delete “that is” L.50-51. Change “which usually evolving to HUS” to “which often evolve into HUS” L.52. Insert “many cases of” between “for” and “non-O157” L.52. The phrase “also, attention should be given to these non-O157 serotypes” is unclear. What kind of attention? L.56. Change “represents” to “causes” L.66. Delete “the” L.67. Change “induces Shiga toxins secretion,” to “may induce Shiga toxin release,” L.78. What is meant by the words “with reflex”? L.82. Insert a comma and “such” after “separation” (i.e., “…separation, such as the Rapid…”) L.83. Insert a comma and “such” after “immunoassay” (i.e., “…immunoassay, such as the Biostar…”) L.104. Change “Klebsiella oxitoca” to “K. oxytoca” (since Klebsiella was spelled out already with pneumoniae and the species is misspelled.) L.104-105. agona, enteritidis, infantis and others are serotypes, not species (they are all serotypes within the species enterica, and by convention should begin with a capital letter and not be italicized. Also, “Salmonella” can be abbreviated as “S.” after it is first spelled out. Hence: “…Salmonella Agona, S. Enteritidis, S. Infantis, S. Newport, S. Typhimurium, …” L.106. Abbreviate “Shigella” as “S.” after it is first spelled out. Hence. “…Shigella boydii, S. flexneri, and S. sonnei)…” L.111. Change “infections” to “infection” L.120-121. Change “…are described in [35].” to “…are also described elsewhere [35].” L.124. Insert “at” between “further” and “4” L.125. Change “Buffered” to “buffered” L.152. Change “E. coli broth” to “EC Broth” and list manufacturer name and location in parentheses. L.168. This reference should be no. 58, not 57. Please review the entire manuscript to make sure there are no more reference errors. L.179. Please spell out UR before abbreviating. L.194-195. Change “…conditions described by [50].” to “conditions as previously described [50].” L.197. Insert “The” before “Vero” and “the” before “gold” in this sentence. L.198. Change “differentiate” to “differentiated” L.198 and throughout the manuscript. Change “stx1” to “stx1” and “stx2” to “stx2” L.205. The meaning of “inter tests” is unclear. Please explain or reword. L.207. Please deleted “toxins” (Stx1 and Stx2 are already understood to be toxins) L.208. Please delete “studied” L.211. Change “…; besides,…” to “…, and…” L.212. Table 1 title. Delete “feature” and change “immunoassays” to “immunoassay”. L.214. Insert a comma after “assay” L.215 and elsewhere in the manuscript. Change “Latex” to “latex” L.216 and elsewhere in the manuscript. Change “Lateral” to “lateral” L.221. Change “assays” to “assay” L.223. Change “Besides” to “In addition” and delete “also” Figures 1 and 2 and their legends. In the figure change “(Number of the strains)” to “(Number of strains).” In the legend, change “True Positive Strains” to “true positive strains”, “True Negative Strains” to “true negative strains”, “Predictive Positive Value” to “positive predictive value”, “Negative Predictive Value” to “negative predictive value”, “Accuracy” to “accuracy”, “Sensitivity” to “sensitivity”, and “Specificity” to “specificity”. Figure 3. In the figure, change “Non detect by Verotoxin” to “Not detected by Verotoxin assay”. In the legend (L.255-256), change “Predictive Positive Value” to “positive predictive value”, “Negative Predictive Value” to “negative predictive value”, “Accuracy” to “accuracy”, “Sensitivity” to “sensitivity”, and “Specificity” to “specificity”. Figure 4. On the x-axis, change the following legends: “DEC non-STEC” to “non-STEC DEC”, “Enterobacterial” to “enterobacterial”, “Microbiota and E. coli non-DEC” to “Microbiota and non-DEC E. coli” L.270. Delete “gene” L272. Change “sorbitol fermenting” to “sorbitol-fermenting”; change “Sorbitol” to “sorbitol” L274. Change “to” to “for” L.284. Change “…tests as …” to “…tests, such as…” L.285. Change “…test as…” to “…tests, such as…” L.287. Delete “Regarding the”, begin the sentence with “Lateral”, and insert a comma after “(LFA)” Change “commercially” to “commercial”Discussion. Will the reagents (mAb and pAb) be made available at a low cost, or only newly developed assays? How do the authors know for sure the cost will be less than commercial assays?
Regarding your comment, we re-evaluate with the Brazilian start-up and we cannot assure now the final costs of the kits. Despite the fact that, we are looking for a more affordable or more attractive kits to the market, in the present moment we only can describe statistical performance of the tests. The company will provide the second step. But we may assume that, it’ll be an option when analyzing the cost benefit issue (bureaucracy, quality, time to obtain the product and final value).
Reviewer 2 Report
General comments:
The authors presented a comprehensive study on the development of immune assays as diagnostic tools as a cost effective way for the detection of Shiga toxin-producing E. coli. The authors brought out the importance of early diagnosis of STEC infection and financial difficulty in some countries in purchasing the commercial kits. The intention of these developed assays were for diagnostic purpose but there was no positive/ negative stools or spiked stool samples included in the study to reflect the true performance of these assays especially when the authors indicate that the assays have the potential to be used as POC testing. The manuscript is well written and it is easy to follow.
Specific comments:
Line 42: “EHEC have gained in importance in the three…” You can drop the word “in”
Line 68: “Besides, the diagnosis allows the rapid notification of outbreaks…” I do not agree with this statement because you can have an increase in number of cases but might not be indicative of an outbreak. Only molecular typing such as PFGE, MLVA and whole genome sequencing can confirm a cluster as an outbreak. You can suspect a potential of an outbreak due to unusual increase in number but only by typing, you cannot come to such conclusion. The authors should modify this sentence.
Result:
I recommend the authors to include photographs of positive and negative controls of the Latex agglutination test and lateral flow assay.
The LAT results in Table 1 is the combined Stx 1 and Stx 2 assay. Where are the LAT results when Stx 1 and Stx 2 were tested individually for calculating the specificity, sensitivity, PPV, NPV and accuracy as indicated in Figures 1 and 2? The performance of the Stx1, and Stx 2 as single target in the LAT should be included in Table 1 as well.
The authors need to confirm all the values pertaining to sensitivity, specificity, PPV, NPV and accuracy.
Figure 3: EIEC 167(48) is written differently in the text on line 250: EIEC (167-48)
Discussion:
Lines 306 to 313: The authors compared the in-house developed assays to the commercial ones in respect to sensitivity and specificity. Were those values based on the kits inserts because those are NOT the values observed in real diagnostic settings and there are few papers published on the true specificities and sensitivities of these tests in a diagnostic laboratory setting with clinical samples. The authors need to clarify the source of the data.
Lines 332 to 337: As point of care testing, the source of the clinical sample will be stools. There is no evaluation in this study that include any positive or negative stool samples or spiking STEC in stool samples with these assays. What will be the extraction method for stools? I see the true value of all these 3 assays but I think consideration has to be made how this can been used in a clinical diagnostic laboratory when diarrhea samples are received. Although the authors have indicated in lines 337 to 339 that knowledge transferred has occurred. There is great value in adding stool samples as part of your validation study especially when the authors present such strong argument in the conclusion.
Author Response
General comments:
The authors presented a comprehensive study on the development of immune assays as diagnostic tools as a cost effective way for the detection of Shiga toxin-producing E. coli. The authors brought out the importance of early diagnosis of STEC infection and financial difficulty in some countries in purchasing the commercial kits. The intention of these developed assays were for diagnostic purpose but there was no positive/ negative stools or spiked stool samples included in the study to reflect the true performance of these assays especially when the authors indicate that the assays have the potential to be used as POC testing. The manuscript is well written and it is easy to follow.
We would like to acknowledge the reviewer comments. We agree with the reviewer, thus we modified the sentence in the in the introduction explaining the goals of our present project. Since in the first version of the manuscript we did not specify the onus of each part of the project. Thus, the following sentence was added in the introduction: “The main project involves two steps: a) searching robust tools for the development of the test; b) will focus on making the use of feces directly, calculating costs and price in the market”.
All corrections were highlighted in the text
Specific comments:
Line 42: “EHEC have gained in importance in the three…” You can drop the word “in”
Sentence was corrected in the manuscript
Line 68: “Besides, the diagnosis allows the rapid notification of outbreaks…” I do not agree with this statement because you can have an increase in number of cases but might not be indicative of an outbreak. Only molecular typing such as PFGE, MLVA and whole genome sequencing can confirm a cluster as an outbreak. You can suspect a potential of an outbreak due to unusual increase in number but only by typing, you cannot come to such conclusion. The authors should modify this sentence.
We agree with the reviewer and we modified the sentence according.
Result:
I recommend the authors to include photographs of positive and negative controls of the Latex agglutination test and lateral flow assay.
As suggested we included the LAT and LFA figures of positive and negative controls.
The LAT results in Table 1 is the combined Stx 1 and Stx 2 assay. Where are the LAT results when Stx 1 and Stx 2 were tested individually for calculating the specificity, sensitivity, PPV, NPV and accuracy as indicated in Figures 1 and 2? The performance of the Stx1, and Stx 2 as single target in the LAT should be included in Table 1 as well.
The individual results of LAT were presented in figures 1 and 2.
The authors need to confirm all the values pertaining to sensitivity, specificity, PPV, NPV and accuracy.
As suggested we checked and confirmed all the statistical performance of the assays.
Figure 3: EIEC 167(48) is written differently in the text on line 250: EIEC (167-48)
This was corrected in the text.
Discussion:
Lines 306 to 313: The authors compared the in-house developed assays to the commercial ones in respect to sensitivity and specificity. Were those values based on the kits inserts because those are NOT the values observed in real diagnostic settings and there are few papers published on the true specificities and sensitivities of these tests in a diagnostic laboratory setting with clinical samples. The authors need to clarify the source of the data.
All data concerning sensitivity and specificity of the assays were taken from datasheet of each company.
Lines 332 to 337: As point of care testing, the source of the clinical sample will be stools. There is no evaluation in this study that include any positive or negative stool samples or spiking STEC in stool samples with these assays. What will be the extraction method for stools? I see the true value of all these 3 assays but I think consideration has to be made how this can been used in a clinical diagnostic laboratory when diarrhea samples are received. Although the authors have indicated in lines 337 to 339 that knowledge transferred has occurred. There is great value in adding stool samples as part of your validation study especially when the authors present such strong argument in the conclusion.
We agree with the reviewer, thus we modified the sentence in the in the introduction explaining the goals of our present project. Since in the first version of the manuscript we did not specify the onus of each part of the project. Thus, the following sentence was added in the introduction: “The main project involves two steps: a) searching robust tools for the development of the test; b) will focus on making the use of feces directly, calculating costs and price in the market”.
Reviewer 3 Report
The paper by Silva et. al. describes the development of a capture a ELISA, a latex agglutination test and a lateral flow assay to detect the production of Shiga toxin in E. coli isolates using previously described polyclonal and monoclonal antibodies. The authors exploit these antibodies to evaluate the sensitivity and specificity of these assays using a panel of 96 E. coli isolates that produce either Stx1 or Stx2 validated by PCR. All three assays showed good levels of sensitivity and specificity using as source of the toxin the strains grown in a specific medium and lysed.
The paper is clearly written and the results and conclusions are correct. This reviewer raises no mayor concerns.
Minor comments:
1) As it is the paper describes the use of these antibodies in three different platforms to detect Stx1 and Stx2 but to truly say these are bonafide diagnostic tools they should have been evaluated using either stool samples or direct detection on the confluent growth zone of plates. Although the manuscript is clear in defining that these could be useful tools to develop new diagnostic assays the title of the manuscript could be tuned down to be more accurate with the conclusions.
2) Maybe the authors could discuss a little more the fact that the antibodies detect all subtypes of Stx1 and Stx2 even though the detection antibodies are monoclonal. Maybe because it detects the A subunit that is more conserved between subtypes?
3) The PPV and NPV really give no additional information in this manuscript because prevalence of the disease is not taken into account.
Author Response
The paper by Silva et. al. describes the development of a capture a ELISA, a latex agglutination test and a lateral flow assay to detect the production of Shiga toxin in E. coli isolates using previously described polyclonal and monoclonal antibodies. The authors exploit these antibodies to evaluate the sensitivity and specificity of these assays using a panel of 96 E. coli isolates that produce either Stx1 or Stx2 validated by PCR. All three assays showed good levels of sensitivity and specificity using as source of the toxin the strains grown in a specific medium and lysed.
The paper is clearly written and the results and conclusions are correct. This reviewer raises no mayor concerns.
We would like to acknowledge the reviewer comments.
Minor comments:
As it is the paper describes the use of these antibodies in three different platforms to detect Stx1 and Stx2 but to truly say these are bonafide diagnostic tools they should have been evaluated using either stool samples or direct detection on the confluent growth zone of plates.We agree with the reviewer, thus we modified the sentence in the in the introduction explaining the goals of our present project. Since in the first version of the manuscript we did not specify the onus of each part of the project. Thus, the following sentence was added in the introduction: “The main project involves two steps: a) searching robust tools for the development of the test; b) will focus on making the use of feces directly, calculating costs and price in the market”.
Although the manuscript is clear in defining that these could be useful tools to develop new diagnostic assays the title of the manuscript could be tuned down to be more accurate with the conclusions.
We changed the title to be more accurate with the conclusions
2) Maybe the authors could discuss a little more the fact that the antibodies detect all subtypes of Stx1 and Stx2 even though the detection antibodies are monoclonal. Maybe because it detects the A subunit that is more conserved between subtypes?
We agree that probably the antibodies recognize the conserved region between subtypes, we cannot discuss a little n=more since we have no evidences.
The PPV and NPV really give no additional information in this manuscript because prevalence of the disease is not taken into account.
We agree in terms, in different areas of low and middle-income regions SHU is sporadic, but in Argentina SHU is endemic, thus we believe for this country the PPV and NPV values of the assays are important to mention.